# MALDI-TOF MS Analysis of Serum Peptidome Patterns in Cervical Cancer

**DOI:** 10.3390/biomedicines11082327

**Published:** 2023-08-21

**Authors:** Phetploy Rungkamoltip, Sittiruk Roytrakul, Raphatphorn Navakanitworakul

**Affiliations:** 1Department of Biomedical Sciences and Biomedical Engineering, Faculty of Medicine, Prince of Songkla University, Songkhla 90110, Thailand; phetploy.run@gmail.com; 2Proteomic Research Laboratory, National Center for Genetic Engineering and Biotechnology, Thailand Science Park, Pathum Thani 12120, Thailand; sittiruk@biotec.or.th

**Keywords:** cervical cancer, signature peptide patterns, matrix-assisted laser desorption/ionization with time-of-flight mass spectrometry

## Abstract

Background: Cervical cancer is the fourth most common cancer among females worldwide. Identifying peptide patterns discriminating healthy individuals from those with diseases has gained interest in the early detection of cancers. Our study aimed to determine signature peptide patterns for cervical cancer screening. Methods: Our study focused on the serum peptidome analysis of 83 healthy women and 139 patients with cervical cancer. All spectra derived from matrix-assisted laser desorption/ionization time-of-flight mass spectrometry were analyzed using FlexAnalysis 3.0 and ClinProTools 2.2 software. Results: In the mass range of 1000–10,000 Da, the total average spectra were represented as the signature pattern. Principal component analysis showed that all the groups were separately distributed. Furthermore, the peaks at *m*/*z* 1466.91, 1898.01, 3159.09, and 4299.40 significantly differed among the investigated groups (Wilcoxon/Kruskal–Wallis test and ANOVA, *p* < 0.001). Conclusions: Laboratory-based rapid mass spectrometry showed that serum peptidome patterns could serve as diagnostic tools for diagnosing cervical cancer; however, verification through larger cohorts and association with clinical data are required, and the use of externally validated samples, such as patients with other types of cancers, should be investigated to validate the specific peptide patterns.

## 1. Introduction

Cervical cancer is the fourth most frequently diagnosed malignancy in females, accounting for an estimated 570,000 new cases and 311,000 deaths globally [1]. In developing countries, cervical cancer cases have significantly decreased due to screening tests. However, the mortality and morbidity rates for cervical cancer remain high, at approximately 85% and 83%, respectively [2,3,4,5]. Moreover, more than 80% of patients with cervical cancer in less developed countries are diagnosed at an advanced stage [6]. Patients with advanced-stage disease are more difficult to treat and receive less effective treatment than those with early-stage disease. Furthermore, the 5-year survival rates also show a negative correlation with the stage of cervical cancer (74.8% for stage I, 58.72% for stage II, and 39.3% for stage III) [7]. The early treatment of cancer, before its metastatic spread to other parts of the body, can significantly increase the chances of survival. To achieve this, adequate screening and diagnostic tests to detect cancers at an early stage play crucial roles [8,9]. However, the current screening and diagnostic tests for detecting cervical cancer are less effective, warranting the identification of novel approaches supporting early detection tests.

Currently, squamous cell carcinoma antigen (SCCA) is used as a biomarker for cervical cancer screening. However, serum SCCA is highly concentrated in patients with squamous cell carcinomas of the esophagus, lungs, head, neck, and anus [10,11]. Therefore, a single biomarker has low sensitivity and specificity. Studies have demonstrated the potential of a panel of biomarker measurements to better describe complex cancer processes and improve cancer detection. Since the first report on the combination of biomarkers published in 2000 [12], several studies have demonstrated the efficacy of this technique in distinguishing patients with cancers and healthy controls. Hill et al. [12] combined s-100 protein, thrombomodulin, myelin basic protein, and neuron-specific enolase, the biomarkers of acute stroke, and revealed that 93% of patients with acute ischemic stroke were diagnosed using the reported panel [12]. Similarly, a combination of biomarkers (CEA, New York esophageal cancer-1 antibody cytokeratin-19 fragment 21-1, hepatocyte growth factor, and CA-125) revealed differential expressions in lung cancer and healthy individuals [13]. The study showed that the combination of biomarkers showed a sensitivity of 49% and a specificity of 96% [13]. Furthermore, a six-biomarker panel including cyclin D1, glutathione S-transferase pi 1, lemur tyrosine kinase 2, hepsin, myosin VI, and fibronectin 1 was used to discriminate between patients with prostate cancer and those with benign prostates. The combination showed higher specificity than the PSA biomarkers alone [14].

Mass spectrometry is an effective tool for detecting protein and peptide expression in biological fluid [15] and has gained interest in developing biomarker panels [16]. Peptides play important roles in several physiological processes, including metabolic products, hormones, and proteolytic enzymes [17]. Proteolytic degradation has been observed in the serum peptidomes, and the analysis of peptidome patterns provides information on pathophysiological processes, especially in cancer [18]. Matrix-assisted laser desorption/ionization time-of-flight mass spectrometry (MALDI-TOF MS) is a powerful approach for observing serum peptide patterns [19,20]. After the co-crystallization of the matrix and sample molecules, the molecules are ionized, resulting in ions traveling directly in the electric field under vacuum conditions before being hit with the detector. The mass spectrum was obtained from the difference between low molecular weight proteins (LMW) and peptides [21]. The serum peptidomes LMW; ≤10 kDa represent the endogenous peptides in both extracellular and intracellular space [22]. Moreover, serum peptide profiles have been utilized to distinguish the presence or absence of malignancy [23].

Cancer development is accompanied by alternative exoprotease activity that generates endogenous peptides. Many researchers have focused on serum peptidomes patterns that may improve the early diagnosis of cancer. Peptidomic pattern analysis has been widely applied in diagnostic tests for gastric, ovarian, hepatic, and breast cancer [24,25,26]. In the ovarian cancer serum, peaks at *m*/*z* 5486, 5643, 5253, 8149, 16,450, 8054, 11,650, 3933, 7773, and 4560 were more highly expressed in ovarian cancer samples than in healthy samples. However, peaks at *m*/*z* 8943, 13,720, 6440, 8142, 6890, 8700, 8575, 13,980, 6369, 4348, and 3260 were lower in ovarian cancer. Therefore, several peptide peaks have been used as biomarker panels to discriminate between patients with ovarian cancer and healthy controls [19]. For instance, peptides with *m*/*z* values of 1310, 2135, 2411, 2585, 3591, 3973, and 4299 were differentially expressed in healthy subjects, patients with pancreatic cancer, and patients with chronic pancreatitis. In contrast, the peaks at *m*/*z* 1209, 1258, 1276, 1448, 1544, 1614, and 1685 were only expressed in healthy subjects [16,27]. Similarly, peptide fingerprints have been used to distinguish between esophageal adenocarcinoma, Barrett’s esophagus, and high-grade dysplasia in healthy subjects. They found that six proteins, including *m*/*z* 1908, 2112, 3158, 3404, 3766, and 4562, were highly expressed in patients with esophageal adenocarcinoma [28]. Pap smears were analyzed via MALDI imaging mass spectrometry for cervical cancer stratification, affecting five peptide peaks at *m*/*z* 2012, 2172, 2339, 3441, and 4740, significantly discriminating between positive and negative cytology [29]. Nevertheless, the serum peptidomic patterns at various stages of cervical cancer have not been explored fully.

Therefore, this study aimed to investigate the peptidome patterns in patients with different stages of cervical cancer and healthy women using MALDI-TOF-MS.

## 2. Methodology

### 2.1. Characteristics of Participants

Signed written informed consent was obtained from all participants, with the approval of the Ethics Committee of the Faculty of Medicine, Prince of Songkla University, Songkhla, Thailand (MED581194S).

The study involved 222 participants, including 83 healthy women, 51 patients with precancer, 24 patients with cervical cancer at stage I, 37 with stage II cervical cancer, and 27 with stage III cervical cancer. Patients with cervical cancer and healthy participants included those visiting the Division of Gynecologic Oncology, Department of Obstetrics and Gynecology, and Division of Blood Bank, Department of Pathology, Songklanagarind Hospital. The age of the participants ranged from 30 to 65 years. Healthy women with no history of cancer or HIV infection were included in this study. The stages of cervical cancer specimens were histopathologically evaluated by a clinician. Patients with cervical cancer were clinically staged according to the revised FIGO 2009 guidelines by the gynecologic and radiation oncologists. Histological classification was based on criteria established by the World Health Organization guidelines. The training set comprised 30 healthy women and 75 patients with cervical cancer. The validation set comprised 117 participants (53 healthy participants and 64 patients with cervical cancer). The characteristics of the participants included in this study are listed in Table 1.

### 2.2. Serum Sample Preparation

Prior to the clinical operation, 5 mL of whole blood was collected from each participant. The collected blood samples were allowed to clot at room temperature for 1 h. For serum collection, whole blood was centrifuged at 2500× *g* for 10 min at 4 °C. Then, all serum was aliquoted and immediately stored at −80 °C until further analysis. Each sample set was analyzed on a distinct day, and a duplicate of each set was assigned to the process in the following week to validate the mass spectrum. The samples at each stage were pooled and quantified using the Lowry assay.

### 2.3. MALDI-TOF MS Processing

Each pooled sample was separated on a Ziptip C18 column (Millipore, Burlington, MA, USA). Ziptip C18 was wetted and equilibrated with 0.1% trifluoroacetic acid (TFA) and 100% acetonitrile (ACN), respectively. The serum was aspirated five times using a Ziptip C18 column to bind peptides with C18 in the resin. Then, Ziptip C18 was eluted with 20 µL 100% ACN.

Serum peptides in the eluted solution were mixed with the MALDI solution, which comprised 10 mg/mL CHCA in 50% ACN containing 0.1%TFA (sample-to-matrix ratio, 1:5). The mixture was then applied to a MALDI-steel target plate (MTP384 ground steel plate; Bruker, Mannheim, Germany). Twenty-four replicates were used to prevent a sample preparation bias. The target plate was placed in an air-conditioned room. After air-drying, mass spectral analysis was performed using an Ultraflex III TOF/TOF (Bruker, Germany) in linear positive mode with a mass range of 1000–10,000 Da and laser shots at 1500 shots/spot. External calibration was operated using standard peptide and protein mixtures, including human Angiotensin II (*m*/*z* 1046), P_14_R (*m*/*z* 1533), human ACTH fragment 18–39 (*m*/*z* 2465), bovine insulin oxidized B chain (*m*/*z* 3465), and bovine insulin (*m*/*z* 5731) (ProteoMass™ peptide and protein MALDI-MS calibration kit; Sigma Aldrich, Burlington, MA, USA). The training and validation sets used the same processing steps for MALDI peptide preparation.

### 2.4. MALDI-TOF MS Data Analysis

All raw mass spectra were obtained using FlexAnalysis 3.0 (Bruker Deltonics GmbH, Bremen, Germany) for spectral processing (smoothing, baseline peak subtraction, MALDI-spectra detection, and spectrum calibration). The total average spectrum, principal component analysis (PCA), and statistical analysis were performed using ClinProTools 2.2 software (Bruker Deltonik GmbH, Bremen, Germany). Different peptides were discriminated using a genetic algorithm (GA), different averages, ANOVA, supervised neural network (SNN), and Wilcoxon/Kruskal–Wallis tests. The two-tailed test was performed to identify significant differences among the investigated groups at a *p*-value of <0.001.

## 3. Results

### 3.1. Peptide Profile Detection

We detected the spectrum signals of the serum samples that were eluted through reverse-phase chromatography-based hydrophobic C18-Ziptip. Peptide patterns from the training and validation sets were analyzed using FlexAnalysis 3.0 and ClinProTool 2.2 software. A validation set was used to validate the training set results. The total average spectra in the range of 1000–10,000 *m*/*z* were acquired for both training and validation sets. The x- and y-axes in the chromatogram represent the mass per charge and intensity, respectively. With the use of MALDI MS, on average up to 184 peaks were found in the validation set, the same as in the training set. Each group exhibited its own peptide pattern. The peptide patterns among the datasets exhibited different *m*/*z* peaks, as shown in Figure 1.

### 3.2. Cluster Analysis

ClinProTools 2.2 software was used to determine whether the peptide patterns of each group were distributed. Different peptide patterns in terms of intensity at a defined mass were created in the PCA view, which applied the mathematical technique for observing the variance within the data. Each sphere represents the mass spectrum of one spot. The distance between each spot describes the similarity of the mass spectra in terms of intensity and mass. Similar peptide patterns were observed in the closed spheres. Colors in PCA expressed the investigated group, including purple (healthy), yellow (precancerous stage), blue (stage I), green (stage II), and red spheres (stage III). As a result, the same color was closed, and the different investigated groups were distant. PCA represents the discrimination peptide patterns not only in cervical cancer and healthy groups but also among different cancer stages, as shown in Figure 2.

### 3.3. The Prominent Expression of Peptide Peaks among the Investigated Groups

Numerous peptide peaks were present in all the investigated groups. In the training set, the average numbers of peaks from healthy participants and patients with precancerous, I, II, and III stages of cervical cancer were 52, 49, 65, 66, and 73, respectively. The average peak numbers for the validation set were 52 (healthy participants), 59 (precancerous), 63 (stage I), 53 (stage II), and 71 (stage III). Different peptide peaks in the peptide mass profiles among the investigated groups were analyzed using a genetic algorithm, supervised neural network (SNN), different averages, an ANOVA *t*-test, and a Wilcoxon/Kruskal–Wallis test, as shown in Table 2. The prominent groups represent high-intensity peaks. A *p*-value of <0.001 was considered statistically significant. Nine peptide peaks, 1466.91, 1488.72, 1741.72, 1898.01, 2044.73, 2307.55, 3159.09, 3242.10, and 4299.40, were significantly different among the groups. Four peptide peaks were observed in patients with cervical cancer (Figure 3);the peak at *m*/*z* 1466.91 was the predominant peak in patients with stage III cervical cancer (Wilcoxon/Kruskal–Wallis test, *p* = 0.0001); the average molecular weight peak at *m*/*z* 1898.01 was significantly predominant in patients with stage I cervical cancer (ANOVA, *p* = 0.000008); the average molecular weight peaks at *m*/*z* 3159.09 from cervical cancer patients with precancerous stage were significantly higher than those in the other investigated groups (ANOVA, *p* = 0.000008); and the average molecular weight peak at *m*/*z* 4299.40 was differentially highly expressed in cervical cancer patients with stage III compared to those in other groups.

## 4. Discussion

Nowadays, most tests for cancer diagnosis are protein-based assays used to discover cancer-associated proteins/peptides. The current use of single biomarkers such as SCCA for cervical cancer diagnosis has been unsuccessful in early phases of cancer due to low expressions. Although other molecules including CA19-9, CEA, and CA125 have been used in cervical cancer detection, these proteins have also been utilized in other cancers [10,11,12,13,14]. It seems likely that no early diagnostic biomarkers with high sensitivity and specificity can be applied as a routine screening assay. A recent study has demonstrated that a combination of biomarkers can enhance sensitivity and specificity for the early diagnosis of cancer [15]. Mass spectrometry-based peptide patterns, as a panel, have been established as an approach for observing combinations of biomarkers. Peptidomic patterns have been widely studied in microorganisms and several diseases, especially gastric, ovarian, hepatic, and breast cancer [24,25,26]. A previous publication reported that using MALDI-TOF MS could discriminate cancer patients from healthy participants by observing peptide patterns. A previous study has identified four MALDI peaks at *m*/*z* 2752, 5866, 6277, and 10,093 to distinguish primary breast cancer from lung adenocarcinoma [30]. For gastric cancer classification, peptides at *m*/*z* 2046, 3179, 1817, 1725, and 1929 were constructed to differentiate patients with gastric cancer from healthy controls. Furthermore, the peptide peaks at *m*/*z* 1741 and 4210 correlated with gastric cancer compared to chronic atrophic gastritis and healthy participants [31,32]. The specific low molecular weight protein profiles in saliva samples were used to discriminate healthy subjects from oral lichen planus, chronic periodontitis, and oral cancer. For instance, the peaks at 5592.26 and 8301.46 Da were significantly increased in oral cancer. The 12,964.55 and 13,279.08 (*m*/*z*) peaks were overexpressed in oral lichen planus [33]. Furthermore, up to 56 difference peaks have been found in patients with and without multiple myeloma, and three significant peaks at *m*/*z* 8131, 11,660, and 22,752 were used to identify multiple myeloma and non-multiple myeloma [15]. Similarly, the MALDI protein patterns were used for discriminating among esophageal adenocarcinoma, Barrett’s esophagus, high grade dysplasia, and healthy patients. It was found that eleven mass spectra had significant difference, and the peaks at *m*/*z* 1908, 2112, 3158, 3404, 3766, and 4562 were overexpressed in patients with esophageal adenocarcinoma [28]. In the range of 700–10,000, the mass spectra were different among biliary tract cancer, benign biliary disease, and healthy controls. The 887, 2903, and 5803 Da had a higher expression in biliary tract cancer than healthy controls [34]. For cervical cancer stratification, the peaks at 3974, 4175, and 5906 Da were specific peptide peaks that distinguished patients with cervical cancer from healthy individuals [35]. Many studies have focused on peptide fingerprints, which may improve cancer screening/diagnosis tests. However, there are few publications on serum peptide patterns at various stages of cervical cancer. In agreement with previous reports, we found different peptide patterns in healthy participants at each stage of cervical cancer. Nine discriminated peptide peaks were differentially expressed in cervical cancer (precancerous, stages I, II, and III) and healthy subjects. Interestingly, peptide patterns may be useful for early cervical cancer diagnosis by creating an in-house database. Moreover, we found four peaks (1466.91, 1898.01, 3159.09, and 4299.40 Da) significantly differentially expressed in cervical cancer.

Previous reports have shown that the 1466 Da peptide is highly expressed in cirrhotic patients with/without hepatocellular carcinoma [36]. The exopeptidase carboxypeptidase Y and P enzyme-digested myoglobin at the C-terminal resulted in a mass peak at 1465 Da, with the DIAA sequence detected via MALDI-TOF MS [37]. Likewise, the 1466 Da protein with sequence DSGEGDFLAEGGGVR was identified as fibrinogen A with an alanine truncation at the N-terminus (degAla-FPA), which was generated from tumor-specific exoprotease activity [24,38]. These data support the hypothesis that serum peptidomes are established by tumor-specific exoprotease activity. Cancer cells may contribute to this unique exoprotease; thus, the peptide patterns in patients with cancer may differ from those in healthy individuals [39,40].

Although the application of MALDI-TOF MS for the diagnosis of cervical cancer was successful, the single laser MALDI approach generates low ion yield, resulting in missing, low-abundant, or hardly ionizable molecules [41]. The low resolution associated with the linear TOF analyzers and the related mass accuracy might limit the discovery of peptide biomarkers. Additionally, MALDI-TOF MS does not allow de novo peptide sequencing in terms of resolution and ability to perform peptide fragmentation and thus identify those specific peptide peaks. According to the limitations of MALDI-TOF MS, peptides with mass spectra linked directly to specific *m*/*z* 1466.91, 1898.01, 3159.09, and 4299.40 are difficult to identify. LC-MS should be used to identify potential peptides specific to cervical cancer. However, LC-MS/MS is time-consuming, expensive, and complicated [33]. In addition, LC-MS/MS with electrospray ionization generates multiple-charged ions that contribute to the loss of singly charged ions, whereas MALDI produces single-charged peptide ions [42,43]. Therefore, MALDI-TOF/TOF enables high-confidence peptide sequence identification.

In summary, our current data are informative and validate peptide signature patterns for the early diagnosis of cervical cancer. Further studies on serum peptide patterns are required to generate an in-house database of diagnostic models. Subsequently, studies involving a large number of patients and healthy participants are required to establish the model. Moreover, the use of externally validated samples, such as those from patients with other types of cancers, should be investigated to validate the specific peptide patterns.

## 5. Conclusions

Our analysis of serum peptide profiles from healthy individuals and patients with various stages of cervical cancer showed that each sample group had its specific pattern of mass signals. Nevertheless, further studies should be carried out, using a larger cohort of patients with cervical cancer and healthy controls and other types of cancer to fully explore the reliability of the serum peptide fingerprint identified in this study as a diagnostic pattern of cervical cancer. Taken together, our findings support the applicability of f MALDI technology combined with bioinformatic tools to facilitate the discovery of cancer-associated peptides and contribute to improved early screening and diagnosis with reproducibility, reduced time consumption, low cost, and high throughput.

## Figures and Tables

**Figure 1 biomedicines-11-02327-f001:**
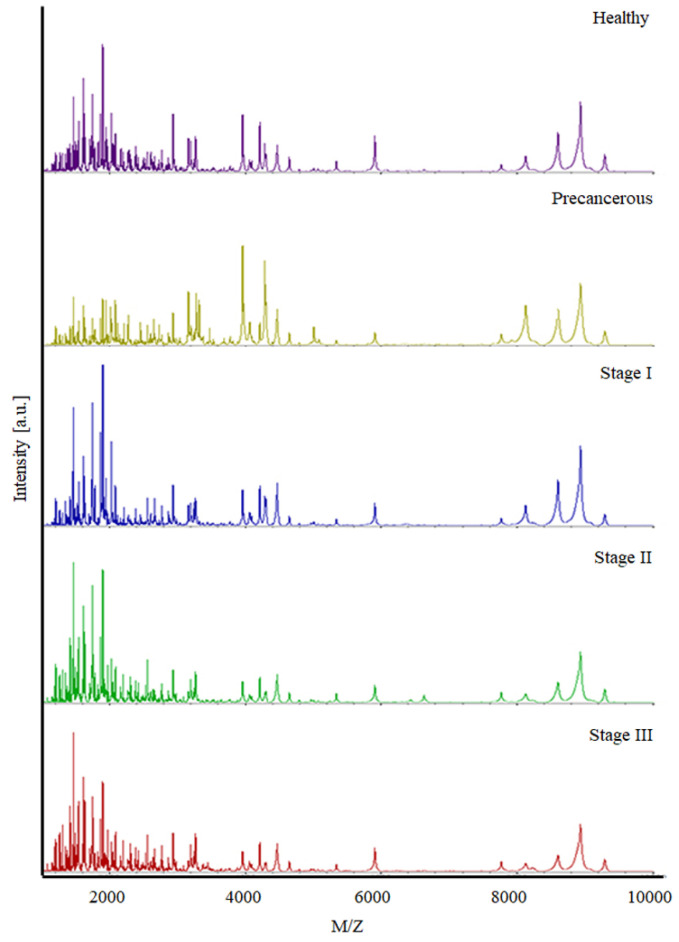
The total average spectra of pooled serum samples in the range of 1000–10,000 *m*/*z*.

**Figure 2 biomedicines-11-02327-f002:**
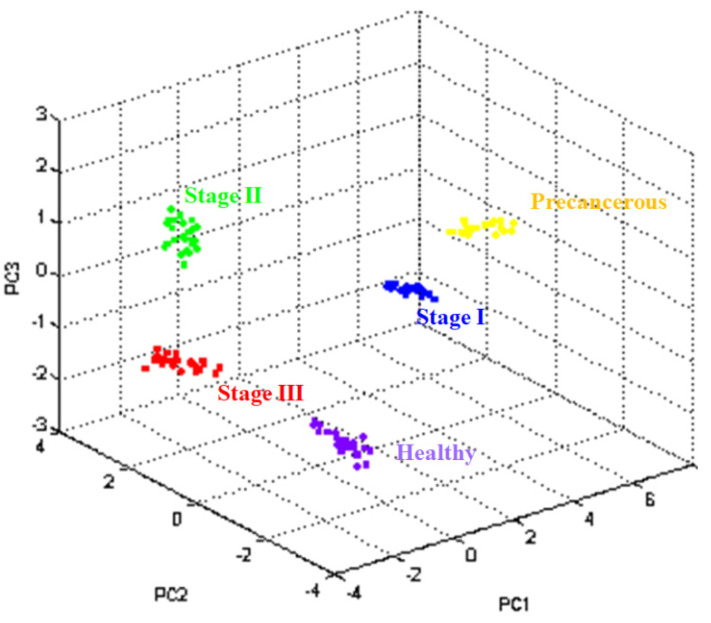
PCA of the pooled sample (preparation using ZipTipC18). The 24 replications (24 spheres) per group were analyzed to observe the precision. The purple, yellow, blue, green, and red spheres represent the mass spectra from healthy controls and patients with precancerous, I, II, and III stages of cervical cancer, respectively.

**Figure 3 biomedicines-11-02327-f003:**
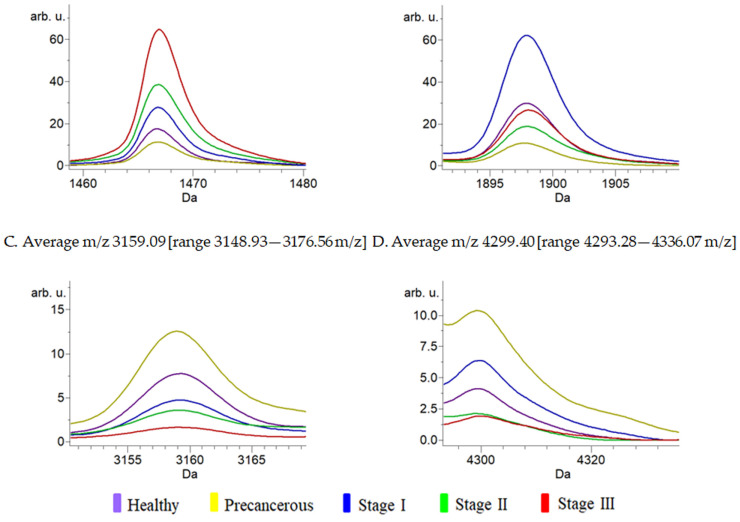
Average peak spectrum that differed significantly among the investigated groups. The average spectra in the range of 1000–10,000 *m*/*z* were analyzed using ANOVA and Wilcoxon/Kruskal–Wallis tests. Serum peptides at *m*/*z* 1898.01 (**A**), 3159.09 (**B**), 4299.40 (**C**), and 1466.91 (**D**) were differently expressed among the investigated groups.

**Table 1 biomedicines-11-02327-t001:** Characteristics of participants from both training and validation datasets.

Sample Group	Pathological Stage	Overall No. of Cases	Training Set		Validation Set	
			No. of Cases	Age (Median/Range)	No. of Cases	Age (Median/Range)
Healthy controls	-	83	30	42/(31–60)	53	42/(30–59)
Patients with cervical cancer	Precancerous	51	30	44/(30–61)	21	45/(30–61)
	Stage I	24	15	45/(32–54)	9	48/(32–65)
	Stage II	37	15	45/(31–65)	22	51/(39–61)
	Stage III	27	15	44/(32–61)	12	52/(43–63)

**Table 2 biomedicines-11-02327-t002:** Different mass signals among pooled stage groups (healthy, precancerous, stage I, II, and III). GA, SNN, different averages, and ANOVA and Wilcoxon/Kruskal–Wallis tests (pWk) were used as the algorithm to observe the significant peaks (*p* < 0.001).

Average Mass	Mass Range (*m*/*z*)	Predominated Group	Statistical Analysis
1466.91	1460.72–1481.63	Stage III	Different average, pWK, *p* = 0.0001
1488.72	1481.63–1498.28	Stage III	GA
1741.72	1436.11–1747.68	Stage I	GA
1898.01	1892.44–1912.26	Stage I	SNN, ANOVA, *p* = 0.000008
2044.73	2038.47–2052.65	Healthy subjects	SNN
2307.55	2300.88–2316.39	Stage II	GA
3159.09	3148.93–3176.56	Precancerous	ANOVA, *p* = 0.000008
3242.1	3231.43–3252.41	Stage II	GA
4299.4	4293.28–4336.07	Stage III	ANOVA, *p*-value = 0.000002

## Data Availability

The data that support the findings of this study are available from the corresponding author upon reasonable request.

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
