# Peer review of "MALDI-TOF MS Analysis of Serum Peptidome Patterns in Cervical Cancer"

_biomedicines, 2023, doi:10.3390/biomedicines11082327_

Round 1
Reviewer 1 Report
The authors aimed at the serum peptidome analysis of 83 healthy women and 139 cervical cancer patients. In their study, the total average spectrums 16 were represented as the signature pattern in mass range 1000-10000 Da. Principal component analysis (PCA) showed that all ex-amines were separately distributed. Furthermore, the peaks at 1466.91, 1898.01, 3159.09, and 4299.40 were significantly differenced among the investigated group. But I don’t think the manuscript is creative enough to be published in this journal. Here are my reasons:
1. Here are some formatting errors, such as Table 1, and what does the last sentence of each figure legend mean (Rungkamoltip P et al., 2019)?
2. The author should give pathological biopsies of some patients with cervical cancer analyzed in this study.
3. Why didn't the authors further analyze the peptides represented by those particular peaks?
4. Which conclusions can be drawn in the article to provide clues to the treatment or prognosis of cervical cancer?
Author Response
|
Comments |
Response to reviewer |
|
Reviewer #1: The authors aimed at the serum peptidome analysis of 83 healthy women and 139 cervical cancer patients. In their study, the total average spectrums 16 were represented as the signature pattern in mass range 1000-10000 Da. Principal component analysis (PCA) showed that all ex-amines were separately distributed. Furthermore, the peaks at 1466.91, 1898.01, 3159.09, and 4299.40 were significantly differenced among the investigated group. But I don’t think the manuscript is creative enough to be published in this journal. Here are my reasons:
|
Thank you for providing us with your valuable suggestions on our manuscript—they have helped us improve the quality of our manuscript. We have prepared a detailed response to your comments and concerns, which we have incorporated into the revised manuscript. The changes have been highlighted for your convenience. Additionally, the revised manuscript has been carefully reviewed by Editage for grammatical and spelling errors. |
|
Here are some formatting errors, such as Table 1, and what does the last sentence of each figure legend mean (Rungkamoltip P et al., 2019)? |
We have changed the format of Table 1 to show the sample information of the datasets. In this study, we focused on patients with different stages of cervical cancer and evaluated their age-matched healthy controls. We apologize for the citation in the Figure legend—it was written mistakenly. We have removed Rungkamoltip P et al., 2019 from all figure legends. |
|
The author should give pathological biopsies of some patients with cervical cancer analyzed in this study. |
Thank you for your comment. Unfortunately, we have limited samples for fresh pathological biopsies due to ethical approval requirements and a lack of funding. However, we believe that our study provides valuable data that can be used for clinical validation in the future. Currently, we are exploring alternative biomarker panels using less or non-invasive clinical samples instead of invasive tissue biopsies. It is important to note that peptide patterns derived from tissue samples can differ from that in those from blood samples. Hence, tissue biopsy may not be consistent with the findings; nevertheless, this is worth investigating. We appreciate your insights and will incorporate in our future studies. |
|
Why didn't the authors further analyze the peptides represented by those particular peaks? |
Thank you for highlighting this. We have planned to characterize the peptides represented by the peaks at m/z 1466.91, 1898.01, 3159.09, and 4299.40 using other MS technique, such as LC-MS/MS. Moreover, as mentioned in our previous answer, we are planning to validate these finding in a larger cohort of patients and healthy individuals and other cancer types. |
|
Which conclusions can be drawn in the article to provide clues to the treatment or prognosis of cervical cancer?
|
We have added the conclusion section in the revised manuscript as utilization of MALDI technology in combination with bioinformatics tools, could facilitate the discovery of cancer-associated peptides and contribute to better early screening and diagnosis in Lines 258–266. |

Reviewer 2 Report
Figure 3 should be further modified. The line color should be bold to more clear show difference. The font of the text should be larger and clear.
